# The contribution of avoidable factors in doubling or halving the odds of hypertension

Jalal Poorolajal[1,2], Younes Mohammadi[1,2], Amin Doosti-Irani[1,3], Saman Khosh-Manesh[1] *

1 Department of Epidemiology, School of Public Health, Hamadan University of Medical Sciences, Hamadan, Iran, 2 Modeling of Noncommunicable Diseases Research Center, Hamadan University of Medical Sciences, Hamadan, Iran, 3 Research Center for Health Sciences, Hamadan University of Medical Sciences, Hamadan, Iran

* khoshmaneshsaman@gmail.com

**Data Availability Statement:** All relevant data are within the paper and its Supporting Information files.

**Funding:** The Vice-chancellor of Research and Technology of the Hamadan University of Medical

## Abstract

### Background

Despite the well-known impact of fruit/vegetable consumption, physical activity, body mass index, waist-hip ratio, fasting blood glucose, and total cholesterol on blood pressure, the amount of exposure to these factors is required to halve or double the odds of hypertension is unknown, but it was investigated in this research.

### Methods

The data used in this study are derived from results of the seventh Iranian national STEPS survey involving 30,542 adults aged 18 years or older. The questionnaire measured non-communicable disease risk factors covered three different levels including behavioral characteristics, physical and biochemical measurements. The level of exposure to factors necessary to reach the odds ratio of 0.5 or odds ratio of 2.0 was obtained from the coefficients of the multiple logistic regression model.

### Results

An OR of 0.5 corresponds to 7 servings of fruit and vegetable consumption daily and burning of 7175 kcal through physical activity at work or recreation daily. An OR of 2.0 corresponds to an increase in body mass index of about 11 kg/m$^2$, an increase in the waist-hip ratio of about 18%, an increase in fasting blood glucose of about 77 mg/dl, and an increase in total cholesterol of about 134 mg/dl.

### Conclusion

The results of this study indicate how much fruit and vegetable and physical activity halve the odds of hypertension and how much increase in body mass index, the waist-hip ratio, fasting blood glucose, and total cholesterol can double the odds of hypertension. Such information may be useful for developing guidelines by policymakers.

Sciences supported this study (No. 9907295282). The funder had no role in study design, data collection and analysis, decision to publish, or preparation of the manuscript.

**Competing interests:** The authors declare that they have no conflicts of interest to declare.

## Introduction

The pressure of blood against the walls of your arteries is known as blood pressure [1]. There are two types of high blood pressure (hypertension): primary (essential) hypertension (with no identifiable cause) and secondary hypertension (caused by an underlying condition). Essential or primary hypertension (high blood pressure) is the most prevalent form of hypertension, accounting for approximately 95% of all cases [2]. Hypertension affects an estimated 1.13 billion people worldwide or nearly 15% of the world's population (1 in 4 men and 1 in 5 women) [3]. Hypertension is a severe but controllable or maybe preventable medical condition that dramatically increases the risk of heart disease, stroke, kidney disease, and vision loss [4].

Hypertension is caused by a complex interaction of genetic, metabolic, and behavioral factors such as fruit and vegetable consumption [5], overweight and obesity [6, 7], blood glucose [8], cholesterol [9], salt intake [10, 11], physical activity [12], alcohol consumption [13], vitamin D deficiency [14], and maybe several other unknown factors. Despite the impact of fruit/vegetable consumption, physical activity, body mass index (BMI), the waist-hip ratio, fasting blood sugar (FBS), and total cholesterol on blood pressure is widely investigated and well-known, however, the extent to which anyone of these factors can reach the odds ratio (OR) of 0.5 or 2.0 has not been thoroughly investigated. Knowing how much more exposure to a factor can halve or double the risk of an outcome of interest is critical in public health policy because it allows for better prioritization and planning of prevention programs. For example, an incremental rise of 20/10 mmHg in blood pressure can double the risk of cardiovascular disease, or participating in 150 minutes of moderate-intensity physical activity per week (or equivalent) will decrease the risk of ischemic heart disease by around 30% and the risk of diabetes by 27% [15].

At the moment, only a few studies have been conducted to determine how much exposure to these well-known factors requires to halve or double the risk of hypertension [16, 17]. According to these studies, an increase in age of about 9.4 years, an increase in BMI of about 10.3 kg/m$^2$, an increase in the waist-to-hip ratio of about 0.5, and an increase in FBS of about 85.8 mg/dl can double the odds of hypertension [16]. In addition, excess weight loss may reduce the risk of hypertension by between 24% and 40% in people who are overweight and by between 40% and 54% in people who are obese [17]. In the current study, we used data of a national screening program (STEPS) to determine the amount of exposure to factors such as fruit/vegetable consumption, physical activity, BMI, the waist-hip ratio, fasting blood glucose, and total cholesterol requires to halve or double the odds of hypertension.

## Methods

The data used in this study are derived from results of the seventh Iranian national survey conducted in 2016, which was a non-communicable disease (NCD) risk factor survey following the WHO STEPwise approach to surveillance (STEPS). This survey was a national project that was conducted for designing and implementing prevention programs against non-communicable diseases. There was no intervention in this study. Therefore, only verbal informed consent was obtained from participants. The study population included 30,542 adults aged 18 years or older. Pregnant women were excluded from the study.

STEPS is the WHO's recommended tool for surveillance of NCDs and their risk factors. We used this tool to collect data and measure NCD risk factors covered three different levels, or 'Steps', of risk factor assessment including (a) questionnaire, (b) physical measurements, and (c) biochemical measurements [18]. Demographic and behavioral information was gathered by a predefined questionnaire in a household setting. Physical measurements were also

done in a household setting. Participants' urine and blood samples were taken in the rural and urban Health Centers and then were sent to a central laboratory.

Basic demographic information included age and sex. Behavioral information included current tobacco use (average number of cigarettes per day), current alcohol consumption (average number of drinks per day/week/month), fruit and vegetable consumption (average servings per day), the median time spent in light/moderate/vigorous physical activity on average per day (minutes) at work, in the household, for transport, and during leisure time. Physical measurements included systolic and diastolic blood pressure (mmHg), height (cm), weight (kg), waist and hip circumference (cm). Biochemical measurements included fasting blood sugar (mg/dl), total cholesterol (mg/dl), HDL-cholesterol (mg/dl) and fasting triglycerides (mg/dl).

A serving size is standardized to represent 80 grams of fruit and vegetable consumption. For example, a banana or an apple of medium size piece; or ½ cup of chopped or cooked fruit; or 1 cup of raw green leafy vegetables such as spinach, salad, etc.; or ½ cup of tomatoes, carrots, pumpkin, corn, cabbage, fresh beans, onion, etc. was considered as one serving unit.

Some people were physically active in all domains (at work, in the household, for transport, and during leisure time), others were active in some domains or were not active in any of the settings. Physical activity was defined as follows [19]:

a. Light-intensity activity is an activity that is classified as <3.5 kcal per minute (2.25 kcal on average), for example, housework, stretching, dancing slowly, leisurely sports (table tennis, playing catch), floating, boating, fishing, etc.

b. Moderate-intensity activity is an activity that is classified as 3.5 to 7.0 kcal per minute (5.25 kcal on average); for example, activities that cause small increases in breathing or heart rate such as carrying light loads, brisk walking, hiking, gardening, fast dancing, swimming, cycling, etc.

c. Vigorous-intensity activity is an activity that is classified as >7 kcal per minute (8.0 kcal on average); for example, activities that cause large increases in breathing or heart rate like such as carrying or lifting heavy loads, digging or construction work, loading furniture, playing football, running, fast swimming, fast cycling, aerobics, etc.

Blood pressure was measured at a sitting position three times allowing the arm to rest for three minutes between each of the readings. The mean of the last two measurements was considered as the participant's blood pressure. Hypertension was defined as systolic and/or diastolic blood pressure ≥140/90 mmHg in adults aged 18 years and over [20]. The patients, who were on medication for hypertension prescribed by a doctor or other health workers, were considered hypertensive even if their blood pressure was normal at the time of measurement.

Waist circumference was measured at the level of the umbilicus. Hip circumference was measured while the measuring tape was horizontal all around the maximum circumference of the buttocks and snug without constricting. The measurements were taken without clothing, that is, directly over the skin in a private area, an area that has been screened off from other individuals within the household. The waist-hip ratio was used as an index of abdominal obesity.

The height was measured in a standing position without footwear (shoes, slippers, sandals, etc.) and headgear (hat, cap, hair bows, comb, ribbons, etc.). For measuring weight, the participants were asked to remove their footwear and take off any heavy belts and empty their pockets. BMI was defined as the weight in kilograms divided by the square of the height in meters and was expressed in units of $kg/m^2$.

To measure fasting blood glucose (FBS) as well as high-density lipoprotein and total cholesterol, the participants were asked to fast for at least 12 hours before blood collection.

The simple and multiple logistic regression model was used to investigate the association between independent variables and hypertension. The OR was considered as the measure of association. The level of exposure to protective factors for hypertension that is necessary to reach the OR of 0.5 was obtained from the following formula: level = ln(0.5)/coefficient. Also, the level of exposure to risk factors for hypertension that is necessary to reach the OR of 2.0 was obtained from the following formula: level = ln(2.0)/coefficient. All analyses were performed at the 2-sided 0.05 significance level (which corresponds to a 95% confidence level) using the Stata software version 16 (StataCorp, College Station, TX, USA).

## Results

Of the 30,542 participants 15,976 (52.3%) were female and 14,566 (47.7%) were male. The mean (SD) age of the participants was 44.50 (16.26) years with a range of 18 to 100 years. The median age was 42 years and the interquartile range (IQR) was 25 years. The prevalence of hypertension was 26.3% (7,842 out of 29,855). It was 27.4% (4,278 out of 15,640) in women and 25.1% (3,564 out of 14,215) in men.

The association between modifiable behavioral, anthropometric, and laboratory factors and hypertension is shown in Table 1. Based on the multiple logistic regression analysis, current daily cigarette smoking, alcohol consumption, dairy consumption, fish consumption, and high-density lipoprotein level had no significant association with hypertension. On the other hand, fruit and vegetable consumption and physical activity had an inversed association with hypertension. Whereas, BMI, waist-hip ratio, levels of blood glucose, and total cholesterol had a positive association with hypertension.

Table 2 shows at what level of some continuous variables, the OR reach a level of 0.5 (indicating the reduction in the odds of developing hypertension) and 2.0 (an increase in the odds of developing hypertension). The interpretation of the coefficient of the multiple regression model is assuming other factors are held constant, for every 1-unit increase in the continuous variable, how much the log odds of hypertension increases. For example, 1 unit increase in fruit/vegetables consumption, the odds of hypertension is exp(-0.09263) = 0.91. Since this is on a continuous scale, a 7 unit increase in fruit/vegetable consumption corresponds to an odds ratio of exp(-0.09263×7) = 0.52. There is also variability associated with this estimate as is represented by the confidence interval of the model coefficient. Based on the above explanation, an OR of 0.5 corresponds to 7 servings of fruit and vegetable consumption daily or burning of 7175 kcal through physical activity at work or recreation daily. On the other hand, an OR of 2.0 corresponds to an increase in BMI of about 11 kg/m$^2$, an increase in the waist-hip ratio of

**Table 1. The association between modifiable behavioral, anthropometric, and laboratory factors and hypertension.**

| Variables | Unadjusted OR (95% CI) | P-value | Adjusted OR (95% CI) | P-value |
|---|---|---|---|---|
| Number of current daily cigarette smoking (cigarette) | 0.994 (0.989, 0.999) | 0.040 | 0.996 (0.990, 1.003) | 0.316 |
| Alcohol consumption (times/week or month) | 0.965 (0.937, 0.995) | 0.023 | 0.982 (0.941, 1.025) | 0.404 |
| Fruit and vegetable consumption (serving/day) | 0.951 (0.935, 0.969) | 0.000 | 0.917 (0.894, 0.940) | 0.000 |
| Dairy consumption (glass/day) | 0.968 (0.942, 0.994) | 0.019 | 0.979 (0.943, 1.017) | 0.283 |
| Fish consumption (serving/week) | 1.001 (0.970, 1.033) | 0.936 | 0.968 (0.927, 1.011) | 0.146 |
| Physical activity at work or recreation (kcal/day) | 0.998 (0.997, 0.999) | 0.000 | 0.998 (0.997, 0.999) | 0.000 |
| Body mass index (kg/m$^2$) | 1.106 (1.100, 1.112) | 0.000 | 1.064 (1.056, 1.071) | 0.000 |
| Waist-hip ratio (%) | 1.054 (1.051, 1.057) | 0.000 | 1.040 (1.036, 1.044) | 0.000 |
| Fasting blood glucose (mg/dl) | 1.012 (1.011, 1.013) | 0.000 | 1.009 (1.008, 1.010) | 0.000 |
| High density lipoprotein (mg/dl) | 0.993 (0.990, 0.996) | 0.000 | 1.003 (0.000, 1.006) | 0.052 |
| Total cholesterol (mg/dl) | 1.007 (1.006, 1.008) | 0.000 | 1.005 (1.004, 1.006) | 0.000 |

**Table 2. Odds ratio estimates of hypertension based on a logarithmic scale using multiple logistic regression adjusted for all variables in the table.**

| Variables | Coef. | SE | z | P-value | 95% CI | | Exposure level † | Effect |
|---|---|---|---|---|---|---|---|---|
| Fruit and vegetable consumption (serving/day) | -0.09263 | 0.01174 | -7.89 | 0.000 | -0.11563 | -0.06962 | 7 | ▼ |
| Physical activity at work or recreation (kcal/day) | -0.00009 | 0.00002 | -4.74 | 0.000 | -0.00013 | -0.00005 | 7175 | ▼ |
| Body mass index (kg/m$^2$) | 0.06155 | 0.00356 | 17.30 | 0.000 | 0.05458 | 0.06853 | 11 | ▲ |
| Waist-hip ratio (%) | 0.03841 | 0.00196 | 19.64 | 0.000 | 0.03458 | 0.04225 | 18 | ▲ |
| Fasting blood glucose (mg/dl) | 0.00895 | 0.00054 | 16.69 | 0.000 | 0.00790 | 0.01000 | 77 | ▲ |
| Total cholesterol (mg/dl) | 0.00517 | 0.00047 | 10.97 | 0.000 | 0.00425 | 0.00610 | 134 | ▲ |
| Constant | -7.57851 | 0.20134 | -37.64 | 0.000 | -7.97312 | -7.18389 | - | |

† The level of exposure to factors necessary to reach the odds ratio of 0.5 (downward green arrows) or odds ratio of 2.0 (upward red arrows).

Formula for protective factors: exposure level = ln(0.5)/Coef.

Formula for risk factors: exposure level = ln(2)/Coef.

about 18%, an increase in FBS of about 77 mg/dl, or an increase in total cholesterol of about 134 mg/dl.

## Discussion

Our findings determined how much amount of exposure to factors such as fruit/vegetable consumption, physical activity, BMI, waist-hip ratio, fasting blood glucose, and total cholesterol can halve or double the odds of hypertension. The amount of exposure reported in this study may be used for the prioritization and planning of prevention programs. However, it is important to remember that risk and protective factors are not separate elements; they interact with each other rather. Therefore, they should be viewed as a whole. Diseases are promoted by risk factors, whereas inhibited by protective factors. The disease will not occur if risk and protective factors are in balance, or if protective factors overcome risk factors. Where risk factors overcome protective factors, however, the disease will occur [21].

The replicated OR of 0.5 or 2.0 reported here for each variable was adjusted for other model variables, in the table. In other words, we used a multiple logistic regression model accounting for all other variables. In multiple logistic regression, each estimated coefficient is the expected change in the log odds of hypertension for a unit increase in the corresponding variables, holding the other variables constant at a certain value.

Our findings indicated that burning of 7,175 kcal through physical activity at work or recreation daily can halve the odds of hypertension. Although this estimate is statistically and theoretically correct, burning such an amount of energy daily may not be practical for many people. Therefore, although physical activity helps lower blood pressure, physical activity alone is not enough to control hypertension. One must consider other factors that help lower blood pressure.

Based on our findings, consumption of every 7 servings/day of fruit and vegetable can halve the odds of hypertension. The evidence suggests that vegetables and fruits reduce blood pressure through various mechanisms. Basic research indicated the important pathways through which fruit and vegetable consumption may contribute to the regulation of blood pressure. A recent review indicated that flavonoids, which are abundant in dietary plants and herbs, play a role in reducing the onset or progression of many cardiovascular diseases, particularly hypertension [22]. Endothelium-dependent microvascular reactivity and plasma nitric oxide increase, while C-reactive protein and E-selectin decrease, in response to a diet rich with high-flavonoid fruits and vegetables [23]. Also, grape polyphenols can potentiate vasorelaxation and

decrease blood pressure and endothelial dysfunction markers [24]. Also, quercetin, a type of flavonoid, was discovered to reduce systolic blood pressure by 3 mmHg [25].

According to our findings, for every 11 unit increase in BMI or 18% increase in the waist-hip ratio, the odds of hypertension doubles. A similar study reported that every 10.3 unit increase in BMI or 50% increase in the waist-hip ratio can double the odds of hypertension [16]. One reason we can give for this difference is that the study was conducted on 7611 people aged over 30 in a single province, while the present study was conducted on 30,542 people aged 18 or older in the whole country. The results of the current study are more valid and robust both in terms of sample size and population diversity.

A systematic review and meta-analysis reported that excess weight loss reduced the possibility of hypertension by between 24% and 40% in overweight people and by between 40% and 54% in obese people [17]. There have been many advances in understanding the pathophysiology of obesity-related hypertension. It is a multifactorial condition in which many possible pathogenic pathways are thought to be involved, including hyperinsulinemia and renin-angiotensin system activation, both of which result in renal tubular reabsorption, sodium accumulation, volume overload, and thus hypertension [26–28]. On the other hand, weight loss can lead to a meaningfully reduced renin-angiotensin-aldosterone system in plasma and adipose tissue, as well as a reduction in insulin resistance, which may contribute to the reduced blood pressure [29, 30].

According to our results, if fasting blood glucose levels rise to 77 mg/dl, the chance of hypertension doubles. Diabetes and hypertension are two conditions that often coexist. A similar study showed that every 85.8 mg/dl increase in blood glucose can double the odds of hypertension [16]. In terms of etiology, the two disorders have a lot in common. Obesity, inflammation, oxidative stress, and insulin resistance are all common metabolic pathways in diabetes and hypertension [31]. Furthermore, metabolic disorders including dyslipidemia, hyperglycemia, and insulin resistance, caused by diabetes, result in a spectrum of physiological changes that induce vascular instability and put patients at risk for atherosclerosis [32]. Diabetes, in other words, increases the risk of hypertension, ischemic heart disease, and stroke in patients because it affects arteries and makes them targets for atherosclerosis [33, 34].

Based on our findings, for every 134 mg/dl increase in total cholesterol, the chance of hypertension doubles. Cholesterol may affect the regulation of blood pressure independently. Evidence showed that cholesterol induces endothelial dysfunction by reducing the bioavailability of endothelium-derived nitric oxide [35, 36]. Endothelium-dependent vasodilation is inversely related to total cholesterol levels [37].

We reported that current daily cigarette smoking had no significant effect on blood pressure. Literature indicated that the effect of smoking on blood pressure is complex with a variety of consequences. While tobacco smoking increases the aortic stiffness and blood pressure acutely, the chronic effects of tobacco smoking on blood pressure and the development of hypertension are uncertain or small [38, 39]. Population-based longitudinal studies are required to indicate whether the acute effect of tobacco smoking on blood pressure may contribute to the progression of chronic hypertension.

Our study was associated with a few limitations and potential biases as follows. First, salt consumption, which is an important factor in hypertension, although measured, was not included in our calculation due to inaccurate measurements. There was the same problem with measuring alcohol consumption. Second, in this survey, we only estimated the association between blood pressure and behavioral factors. The association does not, however, actually imply a cause-and-effect relationship. This might introduce bias in our results. Third, we used a large sample size and performed multiple logistic regression to estimate the pure impact of each factor on the odds of hypertension. However, it is almost impossible to have an 'accurate'

estimate due to complex interactions between several factors affecting hypertension. Despite its limitations, we believe the results of the study are critical for public health policies and setting priorities for prevention programs.

## Conclusion

This study indicated the amount of exposure to fruit/vegetable consumption, physical activity, body mass index, waist-hip ratio, fasting blood glucose, and total cholesterol requires to halve or double the odds of hypertension. Such information is critical in public health policy and may help develop guidelines and better prioritization and planning of prevention programs.

## Supporting information

**S1 Dataset.**
(DTA)

## Acknowledgments

The data used in this study was the results of the seventh Iranian national STEPS survey. The authors thank the National Institute for Health Research (NIHR) of Iran for their support and access to the original data of this survey.

## Ethics approval statement and consent to participate

The Ethics Committee of the Hamadan University of Medical Sciences approved this study (IR.UMSHA.REC.1399.554). The data was obtained from the National STEPS database.

## Author Contributions

**Conceptualization:** Jalal Poorolajal, Younes Mohammadi, Amin Doosti-Irani, Saman Khosh-Manesh.

**Data curation:** Saman Khosh-Manesh.

**Formal analysis:** Jalal Poorolajal, Saman Khosh-Manesh.

**Methodology:** Jalal Poorolajal, Younes Mohammadi, Amin Doosti-Irani, Saman Khosh-Manesh.

**Validation:** Saman Khosh-Manesh.

**Writing – original draft:** Jalal Poorolajal.

**Writing – review & editing:** Jalal Poorolajal, Younes Mohammadi, Amin Doosti-Irani, Saman Khosh-Manesh.

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
