## [Decision Letter · Decision Letter 0]

2 Feb 2022

PONE-D-21-34725The Contribution of Avoidable Factors in Doubling or Halving the Risk of HypertensionPLOS ONE

Dear Dr. Poorolajal,

Thank you for submitting your manuscript to PLOS ONE. After careful consideration, we feel that it has merit but does not fully meet PLOS ONE’s publication criteria as it currently stands. Therefore, we invite you to submit a revised version of the manuscript that addresses the points raised during the review process.

We look forward to receiving your revised manuscript.

Kind regards,

Oliver Chen

Academic Editor

PLOS ONE

Journal Requirements:

“The Vice-chancellor of Research and Technology of the Hamadan University of Medical Sciences supported this study (No. 9907295282). The funder had no role in study design, data collection and analysis, decision to publish, or preparation of the manuscript.”

Please note that funding information should not appear in other areas of your manuscript. We will only publish funding information present in the Funding Statement section of the online submission form.

“The Vice-chancellor of Research and Technology of the Hamadan University of Medical Sciences supported this study (No. 9907295282). The funder had no role in study design, data collection and analysis, decision to publish, or preparation of the manuscript.”

Reviewers' comments:

Reviewer's Responses to Questions

**Comments to the Author**

1. Is the manuscript technically sound, and do the data support the conclusions?

Reviewer #1: No

Reviewer #2: Yes

Reviewer #3: Partly

2. Has the statistical analysis been performed appropriately and rigorously? 

Reviewer #1: No

Reviewer #2: Yes

Reviewer #3: No

3. Have the authors made all data underlying the findings in their manuscript fully available?

Reviewer #1: Yes

Reviewer #2: No

Reviewer #3: Yes

4. Is the manuscript presented in an intelligible fashion and written in standard English?

Reviewer #1: Yes

Reviewer #2: Yes

Reviewer #3: Yes

5. Review Comments to the Author

Reviewer #1: Introduction.

1. For clarity, the authors need to address what different forms of hypertension are.

2. Knowing how much more exposure to a factor can halve or double the risk of an outcome of interest is critical in public health policy because it allows for better prioritization and planning of prevention programs. While it is good to know the impact of each factor on the risk of hypertension, it can be hard to have an accurate estimate due to their complex interactions. For example, fruit and vegetable consumption can affect blood pressure by influencing blood glucose, BMI, etc.

3. “At the moment, only a few studies have been conducted to determine how much exposure to these well-known factors requires to halve or double the risk of hypertension”. Please list out what factors had been examined in 2 previous studies and briefly summarize the findings.

Methods

4. “A serving size is standardized to represent 80 grams of fruit and vegetable consumption.” How were subjects instructed on the serving size? Would they need to weigh the amounts? Did they have a scale at home?

5. “light-intensity activity is an activity that is classified as <3.5 kcal….” Would subjects know how to assess physical activities they performed as light, moderate, or vigorous? The authors need to provide more information.

Results

6. “The mean (SD) age of the participants was 44.50 (6.26) years with a range of 18 to 100 years” Science age is a risk factor for hypertension; more detailed information can be provided, such as median and IQR.

7. The prevalence of hypertension was 26.3% (7,842 out of 30,542). What is the prevalence in men and women?

8. “Based on the multiple logistic regression analysis, current daily cigarette smoking, alcohol consumption, dairy consumption, fish consumption, and high-density lipoprotein level had no significant association with hypertension.” More data on the factors shall be presented since they were reported to associate with blood pressure.

9. “burning of 7175 kcal through physical activity at work or recreation daily.” Will this level of physical activity be practical? If not, the authors may need to use a different OR that can generate an estimate applicable for policy generation.

10. “an increase in FBS of about 77 mg/dl, and an increase in total cholesterol of about 134 mg/dl.” The magnitude of FBS and TC seems not so clinically relevant. What is the OR when FBS is increased by 39 and TC by 67?

Discussion

11. “The evidence suggests that vegetables and fruits reduce blood pressure through various mechanisms.” The authors shall need to list out the mechanisms with citations.

12. “A similar study reported that every 10.3 unit increase in BMI or 50% increase in the waist-hip ratio can double the risk of hypertension “ Since the same group published the study of the reference 15, the authors need to elaborate the differences between 2 study cohorts or the data of the same cohort were used in the analysis.

13. “both of which result in renal tubular reabsorption, sodium accumulation, volume overload, and thus hypertension25-27” After this sentence, it would be great to add the information on how weight loss can decrease the risk of hypertension.

14. The authors need to enrich the discussion section by including more information on the factors identified in the present study and blood pressure in the literature.

Reviewer #2: This paper explored the relationship between fruit/vegetable consumption, physical activity, body mass index, waist-hip-ratio, fasting blood glucose and total cholesterol in Iran and halving or doubling the risk of hypertension. And obtained how much fruit/vegetable and physical activity halve the risk of hypertension and how much increase in body mass index, the waist-hip-ratio, fasting blood glucose, and total cholesterol can double the risk of hypertension. These results may be useful for developing guidelines.

Reviewer #3: This goal of this manuscript was to explore the cut-offs associated with a given odds of having hypertension. Data appeared to have been obtained from a large survey study. The value of such information is noted in such a large scale study population. A few comments to consider.

1) The analysis performed used logistic regression which models the "odds" of an event rather than the "risk" of an event. There is a difference between "odds" and "risk". Revise to match the analysis.

2) The phrase "replicate the odds ratio" is not clear. Consider revising. Possibly something like the target odds ratio?

3) The methods state that "All statistical analysis were performed at a 95% significance level...". This should likely state that all analyses were performed at the 2-sided 0.05 significance level (which corresponds to a 95% confidence level).

4) It may be beneficial to consider rephrasing the analysis that was performed so that is clearer. The goal appear to be, at what level of some continuous variable, does the odds reach a level of 0.5 (indicating the reduction on the odds of having hypertension) and 2.0 (an increase in the odds of having hypertension). The interpretation of the coefficient of the multiple regression model is assuming other factors are held constant, for every 1-unit increase in the continuous variable, the log odds of hypertension increases by xx. For fruit and veggies, a 1 unit increase in fruit/veggie consumption, the odds of hypertension is exp(-0.09263)= 0.91. Since this is on a continuous scale, a 7 unit increase in fruit/veggie consumption corresponds to an odds ratio of exp(-0.09263*7)=0.52. There is also variability associated with this estimate as is represented by the confidence interval of the model coefficient.

5) The discussion section states: "The replicated OR of 0.5 or 2.0 reported here for each variable was adjusted for other model variables, in the table. In other words, we used a multiple logistic regression model accounting for all other variables. Therefore, we were able to measure the pure influence of each variable on blood pressure regardless of the other variables in the model."

However, in multiple logistic regression, each estimated coefficient is the expected change in the log odds of hypertension for a unit increase in the corresponding variables, holding the other variables constant at a certain value. Please revise this sentence with respect to the use of multiple logistic regression.

6) Participants that were already on hypertension medication were coded to be in the "hypertension" group regardless of whether or not medication controlled their hypertension. How many subjects reported being on hypertension medication? This could potentially change the meaning of the modelled odds in that it is the odds of being hypertensive or being medicated for hypertension.

7) To look at factors associated with increased odds of hypertension, what would happen if you did a subgroup analysis of those participants not currently medicated for hypertension? Are results consistent?

6. PLOS authors have the option to publish the peer review history of their article (what does this mean?). If published, this will include your full peer review and any attached files.

Reviewer #1: **Yes: **Oliver Chen

Reviewer #2: No

Reviewer #3: No

---

## [Author Response · Author response to Decision Letter 0]

16 Feb 2022

Reviewer #1: 

Introduction

1. For clarity, the authors need to address what different forms of hypertension are.

Answer: We addressed this issue by adding new information to the first paragraph of the introduction section.

2. Knowing how much more exposure to a factor can halve or double the risk of an outcome of interest is critical in public health policy because it allows for better prioritization and planning of prevention programs. While it is good to know the impact of each factor on the risk of hypertension, it can be hard to have an accurate estimate due to their complex interactions. For example, fruit and vegetable consumption can affect blood pressure by influencing blood glucose, BMI, etc.

Answer: That is right. We can just “estimate” the impact of factors on hypertension but it is not possible to “accurately measure” the impact of a single factor on hypertension. Therefore, we added another sentence to the limitations of the study and explained this issue.

3. “At the moment, only a few studies have been conducted to determine how much exposure to these well-known factors requires to halve or double the risk of hypertension”. Please list out what factors had been examined in 2 previous studies and briefly summarize the findings.

Answer: We added a phrase to the introduction section and briefly summarize the findings.

Methods

4. “A serving size is standardized to represent 80 grams of fruit and vegetable consumption.” How were subjects instructed on the serving size? Would they need to weigh the amounts? Did they have a scale at home?

Answer: This survey was conducted based on the WHO STEPwise guideline. This guideline gave several examples to the participants in order to let them know how much a serving size of fruits and vegetables was. We gave some of these examples in the 4th paragraph of the methods section to clarify this ambiguity.

5. “light-intensity activity is an activity that is classified as <3.5 kcal….” Would subjects know how to assess physical activities they performed as light, moderate, or vigorous? The authors need to provide more information.

Answer: As we explained for the previous question, this survey was performed according to the WHO STEPwise guideline. This guideline gave several examples to the participants in order to let them differentiate between low, moderate, and vigorous intensity physical activity. We gave some of these examples in the 5th paragraph of the methods section to clarify this ambiguity.

Results

6. “The mean (SD) age of the participants was 44.50 (6.26) years with a range of 18 to 100 years” Science age is a risk factor for hypertension; more detailed information can be provided, such as median and IQR.

Answer: We added the median and IQR of age to the first paragraph of the results section.

7. The prevalence of hypertension was 26.3% (7,842 out of 30,542). What is the prevalence in men and women?

Answer: We reported the prevalence of hypertension in men and women separately in the first paragraph of the results section.

8. “Based on the multiple logistic regression analysis, current daily cigarette smoking, alcohol consumption, dairy consumption, fish consumption, and high-density lipoprotein level had no significant association with hypertension.” More data on the factors shall be presented since they were reported to associate with blood pressure.

Answer: We discussed about this issue in details in the discussion section.

9. “burning of 7175 kcal through physical activity at work or recreation daily.” Will this level of physical activity be practical? If not, the authors may need to use a different OR that can generate an estimate applicable for policy generation.

Answer: We gave an explanation in the third paragraph of the discussion section to clarify this issue.

10. “an increase in FBS of about 77 mg/dl, and an increase in total cholesterol of about 134 mg/dl.” The magnitude of FBS and TC seems not so clinically relevant. What is the OR when FBS is increased by 39 and TC by 67?

Answer: We added two formulas to the footnote of the Table 2 to answer this question in another way. The exposure levels of protective and risk factors that we estimated in the last column of the Table 2 are based on OR of 0.5 and 2.0, respectively. However, if someone wants to estimate the exposure levels of protective and risk factors that change the OR (for example) to 0.7 or 1.5, he/she just need to put these figures in the formula and calculate the formula to reach the answer.

Discussion

11. “The evidence suggests that vegetables and fruits reduce blood pressure through various mechanisms.” The authors shall need to list out the mechanisms with citations.

Answer: The references #22, #23, #24, and #25 (which are mentioned in the same paragraph) explain these mechanisms. We moved the above sentence to the beginning of the paragraph to remove this ambiguity.

12. “A similar study reported that every 10.3 unit increase in BMI or 50% increase in the waist-hip ratio can double the risk of hypertension “ Since the same group published the study of the reference 15, the authors need to elaborate the differences between 2 study cohorts or the data of the same cohort were used in the analysis.

Answer: We added additional information to the paragraph in the discussion section and explained the reason for the difference between the two studies.

13. “both of which result in renal tubular reabsorption, sodium accumulation, volume overload, and thus hypertension25-27” After this sentence, it would be great to add the information on how weight loss can decrease the risk of hypertension.

Answer: We added an explanation to the end of this sentence and cited it.

14. The authors need to enrich the discussion section by including more information on the factors identified in the present study and blood pressure in the literature.

Answer: We added some new paragraphs and several phrases to the discussion section and added six new references to the manuscript. The number of references increased from 34 to 40. 

Reviewer #2:

This paper explored the relationship between fruit/vegetable consumption, physical activity, body mass index, waist-hip-ratio, fasting blood glucose and total cholesterol in Iran and halving or doubling the risk of hypertension. And obtained how much fruit/vegetable and physical activity halve the risk of hypertension and how much increase in body mass index, the waist-hip-ratio, fasting blood glucose, and total cholesterol can double the risk of hypertension. These results may be useful for developing guidelines.

Answer: Thank you.

Reviewer #3:

This goal of this manuscript was to explore the cut-offs associated with a given odds of having hypertension. Data appeared to have been obtained from a large survey study. The value of such information is noted in such a large scale study population. A few comments to consider.

1) The analysis performed used logistic regression which models the "odds" of an event rather than the "risk" of an event. There is a difference between "odds" and "risk". Revise to match the analysis.

Answer: We changed the words “risk” related to the results of this study to words “odds”.

2) The phrase "replicate the odds ratio" is not clear. Consider revising. Possibly something like the target odds ratio?

Answer: We paraphrased the whole main text and replaced the word “replicate” with the word “reach”.

3) The methods state that "All statistical analysis were performed at a 95% significance level...". This should likely state that all analyses were performed at the 2-sided 0.05 significance level (which corresponds to a 95% confidence level).

Answer: We replaced the original sentence with the suggested sentence.

4) It may be beneficial to consider rephrasing the analysis that was performed so that is clearer. The goal appear to be, at what level of some continuous variable, does the odds reach a level of 0.5 (indicating the reduction on the odds of having hypertension) and 2.0 (an increase in the odds of having hypertension). The interpretation of the coefficient of the multiple regression model is assuming other factors are held constant, for every 1-unit increase in the continuous variable, the log odds of hypertension increases by xx. For fruit and veggies, a 1 unit increase in fruit/veggie consumption, the odds of hypertension is exp(-0.09263)= 0.91. Since this is on a continuous scale, a 7 unit increase in fruit/veggie consumption corresponds to an odds ratio of exp(-0.09263*7)=0.52. There is also variability associated with this estimate as is represented by the confidence interval of the model coefficient.

Answer: We added the above useful and excellent explanation to end of results section. 

5) The discussion section states: "The replicated OR of 0.5 or 2.0 reported here for each variable was adjusted for other model variables, in the table. In other words, we used a multiple logistic regression model accounting for all other variables. Therefore, we were able to measure the pure influence of each variable on blood pressure regardless of the other variables in the model."

However, in multiple logistic regression, each estimated coefficient is the expected change in the log odds of hypertension for a unit increase in the corresponding variables, holding the other variables constant at a certain value. Please revise this sentence with respect to the use of multiple logistic regression.

Answer: We replaced the original sentence with the suggested sentence.

6) Participants that were already on hypertension medication were coded to be in the "hypertension" group regardless of whether or not medication controlled their hypertension. How many subjects reported being on hypertension medication? This could potentially change the meaning of the modelled odds in that it is the odds of being hypertensive or being medicated for hypertension.

Answer: 22,013 out of 7,842 hypertensive patients used anti-hypertensive medications at the time of study. However, using or not using anti-hypertensive medication had no effect on the results of this study. Because, we considered hypertension as a dichotomous (0, 1) outcome in this study. If considered blood pressure as a continuous variable, using anti-hypertensive medication may affect outcome and hence the results. But it was not the case in this study.

7) To look at factors associated with increased odds of hypertension, what would happen if you did a subgroup analysis of those participants not currently medicated for hypertension? Are results consistent?

Answer: In this study we considered to different approaches to distinguish hypertensive patients from non-hypertensive ones. Measuring the blood pressure (as explained in the methods section) was the first approach. We also assessed participants’ history of using anti-hypertensive medication as an alternative approach to diagnose the hypertensive patients. We considered hypertension as the binomial outcome of interest in this study giving code #1 to the patients and code #0 to non-patients. Taking a history of using anti-hypertensive medication is just a diagnostic approach for increasing the sensitivity of diagnosis of the outcome. Therefore, using or not using anti-hypertensive medication had no effect on the results of this study.

---

## [Decision Letter · Decision Letter 1]

16 Mar 2022

PONE-D-21-34725R1The Contribution of Avoidable Factors in Doubling or Halving the Odds of HypertensionPLOS ONE

Dear Dr. Poorolajal,

Thank you for submitting your manuscript to PLOS ONE. After careful consideration, we feel that it has merit but does not fully meet PLOS ONE’s publication criteria as it currently stands. Therefore, we invite you to submit a revised version of the manuscript that addresses the points raised during the review process.

We look forward to receiving your revised manuscript.

Kind regards,

Oliver Chen

Academic Editor

PLOS ONE

Journal Requirements:

Reviewers' comments:

Reviewer's Responses to Questions

**Comments to the Author**

1. If the authors have adequately addressed your comments raised in a previous round of review and you feel that this manuscript is now acceptable for publication, you may indicate that here to bypass the “Comments to the Author” section, enter your conflict of interest statement in the “Confidential to Editor” section, and submit your "Accept" recommendation.

Reviewer #2: All comments have been addressed

Reviewer #3: All comments have been addressed

2. Is the manuscript technically sound, and do the data support the conclusions?

Reviewer #2: Yes

Reviewer #3: Yes

3. Has the statistical analysis been performed appropriately and rigorously? 

Reviewer #2: Yes

Reviewer #3: Yes

4. Have the authors made all data underlying the findings in their manuscript fully available?

Reviewer #2: No

Reviewer #3: Yes

5. Is the manuscript presented in an intelligible fashion and written in standard English?

Reviewer #2: Yes

Reviewer #3: Yes

6. Review Comments to the Author

Reviewer #2: Specific comments:

1. "Hypertension affects an estimated 1.13 billion people worldwide or nearly 15% of the world's population (1 in 4 men and 1 in 5 women)" in the article does not indicate the reference source.

2. Whether "fasting blood glucose, high-density lipoprotein and total cholesterol" were tested in the field or at a testing facility.

3. Whether people with hypertension were included in the group whose blood pressure was controlled through lifestyle change.

4. Which questionnaire was used for information collection, and whether the reliability and validity of the questionnaire have been verified.

5. It is suggested to update the references.

Reviewer #3: All comments have been addressed and the appropriate adjustments have been made to the manuscript. No further comments.

7. PLOS authors have the option to publish the peer review history of their article (what does this mean?). If published, this will include your full peer review and any attached files.

Reviewer #2: No

Reviewer #3: No

---

## [Author Response · Author response to Decision Letter 1]

22 Mar 2022

Editor’s comments

Please provide additional details regarding participant consent. In the Methods section, please ensure that you have specified (1) whether consent was informed and (2) what type you obtained (for instance, written or verbal). If your study included minors, state whether you obtained consent from parents or guardians. If the need for consent was waived by the ethics committee, please include this information.

Answer: We added the following phrase to the first paragraph in the Methods section. “This survey was a national project that was conducted for designing and implementing prevention programs against non-communicable diseases. There was no intervention in this study. Therefore, only verbal informed consent was obtained from participants. The study population included 30,542 adults aged 18 years or older. Pregnant women were excluded from the study.”

Reviewer #2: Specific comments:

1. "Hypertension affects an estimated 1.13 billion people worldwide or nearly 15% of the world's population (1 in 4 men and 1 in 5 women)" in the article does not indicate the reference source.

Answer: The reference was added.

2. Whether "fasting blood glucose, high-density lipoprotein, and total cholesterol" were tested in the field or at a testing facility.

Answer: Participants’ urine and blood samples were taken in the rural and urban Health Centers and then were sent to a central laboratory. We revised the main text and clarified this issue.

3. Whether people with hypertension were included in the group whose blood pressure was controlled through lifestyle change.

Answer: We applied different approaches to distinguish hypertensive patients from non-hypertensive ones. Measuring the blood pressure (as explained in the Methods section) was the first approach. We also assessed participants’ history of using the antihypertensive medication as an alternative approach to diagnose hypertensive patients. We considered hypertension as the binomial outcome of interest in this study giving code #1 to the patients and code #0 to non-patients. Taking a history of using anti-hypertensive medication is just a diagnostic approach for increasing the sensitivity of diagnosis of the outcome. Therefore, using or not using antihypertensive medication did not affect the results of this study. 

4. Which questionnaire was used for information collection, and whether the reliability and validity of the questionnaire have been verified.

Answer: STEPS is the WHO's recommended tool for surveillance of NCDs and their risk factors. We used this tool to collect data and measure NCD risk factors. We added an explanation to the methods section to clarify this issue.

5. It is suggested to update the references.

Answer:

Vallance 1989 was replaced with Maruhashi 2013.

Clarkson 1996 was replaced with Petrie 2018.

Creager 1997 was replaced with Chen 2018.

Hayakawa 1999 was replaced with Hua 2019.

Beckman 2002 was replaced with Poznyak 2020.

Whelton 2002 was replaced with Tasnim 2020.

Chobanian 2003 was replaced with Whelton 2018.

Rice-Evans 2003 was replaced with Maaliki 2019.

Accordingly, all references are from 2005 and beyond, and about half of them are from the last 5 years.

Reviewer #3

All comments have been addressed and the appropriate adjustments have been made to the manuscript. No further comments.

Answer: Thank you.

---

## [Editor Report · Decision Letter 2]

25 Mar 2022

The Contribution of Avoidable Factors in Doubling or Halving the Odds of Hypertension

PONE-D-21-34725R2

Dear Dr. Poorolajal,

We’re pleased to inform you that your manuscript has been judged scientifically suitable for publication and will be formally accepted for publication once it meets all outstanding technical requirements.

Kind regards,

Oliver Chen

Academic Editor

PLOS ONE
---

## [Editor Report · Acceptance letter]

29 Mar 2022

PONE-D-21-34725R2 

The Contribution of Avoidable Factors in Doubling or Halving the Odds of Hypertension 

Dear Dr. Poorolajal:

I'm pleased to inform you that your manuscript has been deemed suitable for publication in PLOS ONE. Congratulations! Your manuscript is now with our production department. 

Kind regards, 

on behalf of

Dr. Oliver Chen 

Academic Editor

PLOS ONE